# Data-Augmented Manifold Learning Thermography for Defect Detection and Evaluation of Polymer Composites

**DOI:** 10.3390/polym15010173

**Published:** 2022-12-29

**Authors:** Kaixin Liu, Fumin Wang, Yuxiang He, Yi Liu, Jianguo Yang, Yuan Yao

**Affiliations:** 1Institute of Process Equipment and Control Engineering, Zhejiang University of Technology, Hangzhou 310023, China; 2Maynooth International Engineering College, Fuzhou University, Fuzhou 350108, China; 3Department of Chemical Engineering, National Tsing Hua University, Hsinchu 300044, Taiwan

**Keywords:** non-destructive evaluation, deep learning, thermographic data analysis, generative adversarial network, defect detection, manifold learning

## Abstract

Infrared thermography techniques with thermographic data analysis have been widely applied to non-destructive tests and evaluations of subsurface defects in practical composite materials. However, the performance of these methods is still restricted by limited informative images and difficulties in feature extraction caused by inhomogeneous backgrounds and noise. In this work, a novel generative manifold learning thermography (GMLT) is proposed for defect detection and the evaluation of composites. Specifically, the spectral normalized generative adversarial networks serve as an image augmentation strategy to learn the thermal image distribution, thereby generating virtual images to enrich the dataset. Subsequently, the manifold learning method is employed for the unsupervised dimensionality reduction in all images. Finally, the partial least squares regression is presented to extract the explicit mapping of manifold learning for defect visualization. Moreover, probability density maps and quantitative metrics are proposed to evaluate and explain the obtained defect detection performance. Experimental results on carbon fiber-reinforced polymers demonstrate the superiorities of GMLT, compared with other methods.

## 1. Introduction

As one of the advanced composite materials, the demand for carbon fiber-reinforced polymer (CFRP) in new energy, equipment manufacturing, and other fields is growing rapidly [1]. Research on its nondestructive evaluation (NDE) and structural health monitoring has become a necessary and interesting topic. During the manufacturing and long-term service of CFRP products, defects are inevitably generated inside the materials. The unknown sizes, shapes, locations, and physical properties of the defects make the NDE study of CFRP a challenge. Infrared thermography (IRT) [2], as a classical non-destructive test technique, has gained wide interest in CFRP quality assessment and cultural heritage restoration [3,4,5,6,7]. Although IRT offers several advantages in terms of detection efficiency, its direct performance does not provide satisfactory results. In most practical investigations, inhomogeneous backgrounds and noise mask the objects in IRT-recorded thermal images. Moreover, the manual inspection of each thermal image is time-consuming and laborious, and the detected results are ambiguous. Therefore, processing and analyzing thermographic data has become an important procedure to enhance the accuracy and scope of IRT applications, which has emerged as a growing concern for IRT researchers.

Machine learning methods have been widely utilized in various fields over the last two decades [8,9,10], including the processing and analysis of thermograms in IRT-based defect detection. Typical thermographic data analysis methods include principal component thermography (PCT) [11], sparse PCT [12], thermographic sequence reconstruction (TSR) [13], blind source separation [14], autoencoder [15], and convolutional neural networks [16]. It is worth mentioning that manifold learning [17,18,19] methods recently introduced into the field of thermography show promising defect detection performance [20,21]. Nevertheless, it is difficult to obtain an accurate and explicit mapping from the input data manifold to the output embedding in the manifold thermography method because of the serious collinearity between the thermal images. Additionally, due to the limited thermal images, the performance of most thermogram analysis models aforementioned is restricted, and there is still potential for improvement.

Obtaining informative data is one feasible way to improve the performance of machine learning algorithms, especially in tasks where short or small data problems exist [22]. In cases where data acquisition is costly or even impossible, data augmentation methods are an alternative way to artificially increase the training dataset. For small data problems of tabular or numerical type, data augmentation approaches can be divided into two main subclasses: columnar methods and row-wise methods. Among them, the input doubling method is a class of representative methods [23,24]. However, if we talk about image data, the generative adversarial networks (GAN) family has become promising as a notable data augmentation method in recent years. It has shown high efficiency in enlarging image datasets and has been increasingly applied in various fields [25,26,27]. Recently, GAN employment in IRT has been reported. Guei et al. [28] presented a deep learning framework using GAN-based augmented data for IR face image super-resolution improvement and showed good results. Liu et al. [29] proposed a generative PCT (GPCT) method to improve the PCT-based defect detection performance in composites. However, to the best of our knowledge, the fusion framework of GAN-derived methods with manifold learning methods with nonlinear feature extraction has been rarely investigated, which may be a breakthrough to enhance the accuracy and reliability of IRT defect detection and evaluation.

The aim of this work is to facilitate the success of thermography-based defect detection through data augmentation and analysis, achieving an accurate inspection of the geometric properties of subsurface defects in the specimen. The main contributions can be summarized as follows:A generative manifold learning thermography (GMLT) method is proposed for defect detection and the evaluation of polymer composites.The spectral normalization generative adversarial network (SNGAN) is designed as a thermogram augmenter which enlarges the dataset; the isometric feature mapping (ISOMAP) manifold learning is adopted to learn the intrinsic geometric structure of the nonlinear thermographic data; and the partial least squares regression (PLSR) latent variable method is proposed for visualizing defects.A quantified defect–background separation index is developed for the performance evaluation of different methods. Experiments on CFRP specimens demonstrate the advantages of the proposed GMLT method. The probability density plot is used for model interpretation.

## 2. Thermal Image Data Structures

In IRT defect detection, the experimental system is an indispensable experiment foundation, which provides the thermal images directly for subsequent thermographic data analysis methods. Here, the main components and functions of the IRT system are briefly introduced so that the readers can better understand its working principle and refer to it in practice. Figure 1 shows the general infrared system that contains three components: an excitation heat source comprising one or two flash lamps, an infrared camera, and a computer.

The operation process of each component of the IRT system is described briefly. First, the flash lamp is used as the heat source to emit thermal pulses for heating the specimen. Then, an infrared camera is used to record the surface temperature pattern of the test specimen during the cooling stage. Finally, the captured thermal images are stored on a computer for subsequent analysis. As shown in Figure 2, the temperature pattern is recorded as a $(N_{x}\times N_{y})\times N_{t}$ hyper-image, which may be viewed as a series of thermograms recorded at *N_t_* sampling instants. Each thermogram is $N_{x}\times N_{y}$ in size. For each pixel, the color indicates the surface temperature at the corresponding location. Each pixel corresponds to a temperature change signal measured at the *N_t_*-th time instant. Moreover, as shown in Figure 2, for most of the thermal image analysis methods, it is necessary to convert the three-dimensional (3D) thermographic data into a 2D matrix by stretching each thermogram into a 1D vector [11,12,13,14].

## 3. Methodology

In this section, a deep learning-based GMLT framework is established to facilitate IRT defect assessment. As shown in Figure 3, the framework consists of four parts, including SNGAN-based thermogram augmentation, ISOMAP manifold learning, visualization of defect patterns using PLSR, and quantitative indicators for method evaluation.

### 3.1. SNGAN-Based Thermogram Augmentation

Data augmentation has become one primary method of saving data collection costs to solve small data problems [28,29,30]. For the thermographic data analysis, a critical factor restricting the performance of the thermography method is the limited number of thermal images. Therefore, it is a natural idea to adopt the data augmentation strategy to solve the challenge of the insufficient amount of data faced by thermography methods. Among the data augmentation techniques, thanks to the clever game structure design, GAN and its derivatives perform well in several application scenarios and provide promising solutions [26,27,28,29]. In this work, SNGAN [25], as an improved GAN algorithm, is used as a data augmentation tool to augment the dataset and enrich the diversity of the data.

The SNGAN contains two deep neural networks, one is a generator and the other is a discriminator. Compared with the GAN, the other structures of SNGAN remain unchanged, but the weight matrix of the discriminator is spectrally normalized [25]. This structure improvement allows SNGAN to obtain good capabilities without the intensive adjustment of the hyperparameters, whereas the implementation is simple, and the additional calculation is small. Specifically, spectral normalization is performed on the weight matrix $W_{i}$ of each layer of the discriminator [25]:(1)$${\hat{W}}_{i}=\frac{W_{i}}{\sigma (W_{i})}$$
where ${\hat{W}}_{i}$ is the weight matrix after the spectral normalization of each layer; $\sigma (W_{i})$ is the spectral parameterization of the weight matrix (i.e., *L*_2_ matrix norm of $W_{i}$). At this time, all layers of the discriminator satisfy the 1-Lipshcitz limit, and the entire discriminator meets the 1-Lipshcitz limit. In this way, SNGAN solves the legacy problem of the Wasserstein GAN and has better data generation capability than the general GAN [25].

The SNGAN-based thermography augmentation architecture is shown in Figure 4, which is used to generate images similar to the original thermal image distribution. In this model, the generator *G* contains one fully connected layer and four deconvolutional layers, and the discriminator *D* contains one fully connected layer and three convolutional layers. All the convolution kernels are set to 3 × 3. All the convolution steps of the whole model are 2, except for the last deconvolution layer of *G* where the convolution step is 1. Figure 4 shows the structure of each layer and the number of neurons set. It should be emphasized that the ReLU and tanh activation functions are used in the *G* network, while the leaky ReLU activation function is used in the *D* network. More importantly, compared to the *G* network, the *D* network performs spectral normalization at each layer in addition to the use of batch normalization.

### 3.2. ISOMAP-Based Manifold Learning Thermography

As a manifold learning algorithm, the ISOMAP [17] has been applied to the domain of thermography and developed as a manifold learning thermography (MLT) method for the detection of CFRP [20]. It distinguishes the inhomogeneous background, noise, and defect features in thermal images, enabling the detection of defects effectively. However, like most thermal image data analysis methods, the MLT method also faces the challenge of thermogram shortage, which limits its defect detection performance.

In this work, the GMLT approach is proposed to address the challenge of insufficient thermographic data. Section 3.1 presents the details and procedures of the thermogram augmentation. This section introduces how to build the ISOMAP manifold architecture for feature extraction in the GMLT framework. It is assumed that the IRT experiment obtains a 3D thermographic dataset of $(N_{x}\times N_{y})\times N_{t}$, representing *N_t_* frames of $N_{x}\times N_{y}$ thermal images. In the beginning, the 3D thermographic dataset is used as input data, and the SNGAN data augmentation model is trained. The generated data have almost the same distribution as the real thermographic data when the model converges. Finally, *N_g_* simulated thermal images are obtained. Similar to the original data, they are viewed as a $(N_{x}\times N_{y})\times N_{g}$ 3D matrix.

Then, the original IRT data and the SNGAN-generated data are merged and preprocessed. Specifically, the (Nx×Ny)×Nt dataset is merged with the (Nx×Ny)×Ng dataset into a 3D-integrated dataset with dimensions Nx×Ny×(Nt+Ng). The merging process is to attach the generated thermal images to the original dataset. As manifold learning often deals with the eigendecomposition of a matrix, the 3D Nx×Ny×(Nt+Ng) integrated dataset is transformed into an (Nx×Ny)×(Nt+Ng) 2D **X**. Each column of **X** is a one-dimension expansion of a single thermal image, and each row is the temperature response at a single pixel location. To reduce the computational effort and efficiently extract features, a normalization procedure is performed on the matrix X=[x1,x2,⋯,xNt+Ng]∈ℝ(Nx×Ny). This means that each column is subtracted from its mean and divided by its standard deviation. For simplicity, the normalized matrix is still denoted by **X**.

Next, the ISOMAP method is implemented to solve the low-dimensional embedding **Y** of **X**. Let each column **x***_i_* of **X** be a sample. The *k*-nearest neighborhood algorithm is used to construct the neighborhood graph *H*. After this, the Euclidean distance $d_{X}(x_{i},x_{j})$ between two adjacent sample points, $x_{i}$ and $x_{j}$, is calculated as the edge weight. Later, the ISOMAP utilizes Dijkstra’s algorithm to find the shortest path and calculates the shortest path distance $d_{H}(x_{i},x_{j})$ to estimate the geodesic distance between all pairs of points on the manifold [19,20]. Specifically, the ISOMAP initializes $d_{H}(x_{i},x_{j})=d_{X}(x_{i},x_{j})$ if **x***_i_* and **x***_j_* are linked by an edge; otherwise, $d_{H}(x_{i},x_{j})=\infty$. For each value of *r* = 1, 2, ∞, $N_{t}+N_{g}$, in turn, replace all the entries $d_{H}(x_{i},x_{j})$ by $\min\{d_{H}(x_{i},x_{j}),d_{H}(x_{i},x_{r})+d_{H}(x_{r},x_{j})\}$. The matrix of final values $D_{H}=d_{H}(x_{i},x_{j})$ contains the shortest path distances between all pairs of points in *H*.

Then, multidimensional scaling is applied to the matrix of distances $D_{H}=d_{H}(x_{i},x_{j})$ to construct the low-dimensional coordinates **Y**. Let the squared distances matrix $S_{i,j}={\left(D_{H}\right)}_{i,j}^{2}\;$ and the centering matrix **H** be calculated as follows:(2)$$H_{i,j}=\delta_{i,j}-\frac{1}{\left(Nt+Ng\right)}I$$
where $\delta_{ij}=\left\{\begin{array}{ll} 0 & i=j\\ 1 & i\neq j \end{array}\right.$ and **I** are an identity matrix. 

Define the operator **τ** as follows:(3)$$\tau_{H}=-\frac{\;HSH\;}{2}$$

After the eigenvalue decomposition of the matrix $\tau_{H}$, the first *d* eigenvalues (in descending order) ${\left\{\lambda_{i}\right\}}_{i=1}^{d}$ and the corresponding *d* eigenvectors ${\left\{\tau_{i}\right\}}_{i=1}^{d}$ are obtained. Finally, the ISOMAP dimensionality reduction result, i.e., the embedding matrix **Y** of the input data **X**, is calculated, and $Y={\left[\sqrt{\lambda_{1}}\tau_{1},\sqrt{\lambda_{2}}\tau_{2},\ldots ,\sqrt{\lambda_{d}}\tau_{d}\right]}^{T}$.

### 3.3. Defect Patterns Visualization Using PLSR

The main drawback of the manifold learning approach is that there is no explicit mapping from the input data manifold to the output embedding. When the ISOMAP method is used for IRT data analysis, the low-dimensional embedding results do not provide direct visual convenience for defect detection. Previously, the MLT method employed ordinary least squares regression (OLSR) in finding the explicit mappings [20]. However, there is serious collinearity in the thermographic data, resulting in unstable solutions and hence affecting the visualization results.

Here, PLSR is used to solve the explicit mapping and give the procedure for defect visualization [31]. One main reason is that it is competent for analysis scenarios where the number of samples is less than the number of variables, and there are significant multiple correlations in the independent variables. In this task, the PLSR model takes the column vectors of the **X** and **Y** as samples and each row as a dimension. The column vectors of matrix **Y** are normalized before the analysis. The multivariate model of PLSR is as follows:(4)$$X=VP^{T}+E$$
(5)$$Y=UQ^{T}+F$$
where **X** is the $\left(N_{x}\times N_{y})\times (N_{t}+N_{g}\right)$ thermographic data matrix; **Y** is the $(N_{x}\times N_{y})\times d$ ISOMAP embedding matrix; **V** and **U** are the score matrices obtained after applying principal component analysis (PCA) for **X** and **Y**, respectively; **P** and **Q** are the load matrices obtained after applying PCA for **X** and **Y**, respectively; and the matrices **E** and **F** are the corresponding error terms.

The relationship between **X** and **Y** cannot be established by directly using (4) or (5). Regression analysis is performed on the score matrices **V** and **U** based on the correlation of the column vectors to obtain the transformation matrix **B**:(6)U=VB
where $B=(V^{T}V^{-1})V^{T}U$.

The transformation matrix **B** describes how matrix **X** is mapped by ISOMAP to the embedding matrix **Y**, which can be used as a display mapping for ISOMAP dimensionality reduction. To visually defect defects, each column vector (noted as the indicator vector, IV) of matrix **B** is reconstructed into a two-dimensional matrix of size $N_{x}\times N_{y}$, and then visualized as an IV thermogram using a Jet color mapping algorithm with high contrast that can effectively highlight image details. Eventually, *d* IV images (*d* << ($N_{t}+N_{g}$)) with the geometric properties (location, shape, and size) of the defects can be observed, thus contributing to the efficiency and accuracy of defect detection by IRT.

### 3.4. Quantitative Indicators for Method Evaluation

The main goal of thermographic data analysis methods is to extract defect features from the thermographic data. Indicators for evaluating the performance of different methods are developed for quantifying the extracted defect information. The thermographic defect detection problem discussed in this work is an unsupervised learning task. Therefore, indicators designed for supervised learning are generally not available. There are few existing indicators that are suited to evaluating the performance of unsupervised thermogram processing methods. Among them, the signal-to-noise ratio (SNR) [32] is a popular and widely used criterion which reflects the ratio between defect information and noise in the analysis results. It is calculated as follows:(7)$$\mathrm{SNR}=\frac{M_{\mathrm{def}}-M_{\mathrm{in}}}{\sigma_{\mathrm{in}}}$$
where $M_{\mathrm{def}}$ is the average value of pixel values in the defective area, $M_{\mathrm{in}}$ is the average value of pixel values in the intact area, and $\sigma_{\mathrm{in}}$ is the standard deviation value of pixel values in the intact area. The SNR is a dimensionless indicator. In situations where the backgrounds are homogeneous, a large SNR value indicates a better defect detection performance.

However, when the backgrounds are inhomogeneous, the SNR indicator often provides results against human experiences. The reason is that this index reflects the global signal-to-noise ratio, while human beings identify defects based on the local contracts between the defect regions and their surroundings. To solve this, this work proposes a defect-background separation (DBS) indicator as a complement to SNR. It is calculated as follows:(8)$$\mathrm{DBS}=\frac{{\left|M_{\mathrm{def}\;}-M_{\mathrm{in}\;}\right|}^{2}}{\sigma_{\mathrm{def}}{}^{2}+\sigma_{\mathrm{in}}{}^{2}}$$
where the pixels of the defective area are recorded as one class, and the pixels of the intact areas are denoted as the other. $M_{\mathrm{def}}$ and $\sigma_{\mathrm{def}}{}^{2}$ are the average value and variance value of the pixels in the defective area of the thermogram, respectively. $M_{\mathrm{in}}$ and $\sigma_{\mathrm{in}}{}^{2}$ are the average value and variance value of the pixels in the intact area, respectively. 

The DBS index evaluates the thermogram analysis methods from the perspective of signal separation. Specifically, the DBS index quantifies the degree of separation of information between defects and intact areas in the extracted features, which is related to the ease of identifying defects with the naked eye and helps determine the shape and location of defects. A larger DBS value indicates a larger inter-class distance and a smaller intra-class distance, which means the defect is more separable from the intact area. Therefore, a larger DBS indicates better defect identification results.

## 4. Experiments and Results

### 4.1. Thermogram Dataset Preparation

The shapes of defects in real industrial products composed of composite materials are usually irregular and diverse. It is difficult to evaluate these products because the unknown nature of its defects makes it impossible to judge the accuracy of the evaluation results. In this experiment, a CFRP specimen was fabricated using the resin transfer molding process [20]. The 20 carbon fiber sheets and epoxy resin used to form the CFRP specimen had a thickness of roughly 1 cm and a planar size of roughly 18 cm × 18 cm. Three Teflon strips were diamond, circle, and trapezoid., respectively. Each of them had an area of about 3 cm^2^ and were inserted into the carbon fiber sheets prior to the injection of thermosetting resin into the fiber preforms that had been placed in a closed mold. The deepest defect, which is diamond-shaped, is located under three layers of carbon fiber sheets, the circular defect is located under two layers of fiber sheets, and the trapezoidal defect is located under a single layer. A diagram of the CFRP specimen with defects is shown in Figure 5, where each defect is located in a different plane and at a different depth. This distribution information is used to judge the results of thermographic data analysis methods including GMLT.

After the specimen is prepared, pulse thermography (PT) is used to detect internal defects. PT [15,20] is one of the commonly used IRT techniques to detect defects. The configuration of the PT system is shown in Figure 1, with a 3200 W·s flash lamp to heat the specimen by emitting thermal pulses. The parameters of the infrared camera are set as follows: model, TAS-G100EXD, NEC type; sensor, uncooled microbolometer thermal focal plane; resolution, 320 × 240 pixels; wavelength, 8–14 μm; temperature range, –40 to 1500 °C; sampling rate, 30 frames/second; and experimental layout, reflection mode. Finally, a total of 90 thermal images were obtained in three seconds. The study subsequently focused only on the 308 × 212 pixels region of interest in each thermogram. Figure 5 shows three thermal images recorded at different sampling instants, where the heavily inhomogeneous background masks the defect information.

### 4.2. Defect Detection Results of CFRP

In this case, TSR, PCT, and MLT methods were performed to evaluate the subsurface defects of CFRP specimens for comparison with GMLT. TSR, as a signal reconstruction method, improves the spatiotemporal resolution of thermographic data by filtering the data in the time direction. Figure 6 shows that the four thermograms reconstructed by the TSR method appear to have the same visual state. Compared to the original thermograms, the TSR reveals the rough location of the defects. However, TSR only applies a polynomial filter in the time dimension; it does not take advantage of the large amount of spatial information contained in the thermogram. Consequently, it is difficult to determine the shape of the defect, and there is still a certain uneven background and noise in the images.

As a representative data dimensionality reduction method, PCA and its extended algorithm have been applied in thermography because of its ability to reduce noise and simple implementation [11]. As shown in Figure 7, six principal component (PCs) images of the PCT method show three types of information including uneven background, noise, and defects. Compared to the original thermograms and the TSR method analysis results, the PCT method obtains better performance for defect detection. PC1 mainly contains an uneven background, while PC3 and PC4 show prefabricated defects, the location of which is clear. However, the contrast between defects and intact areas in the PCT analysis results is not conspicuous for defect identification.

Figure 8 shows the ISOMAP-based MLT approach, which uses the PLSR method to visualize defects. Notice that the MLT method here is an improved version compared with the one used in ref. [20]. The target dimension is set to 6, and the number of nearest neighbors is set to 5. Its results of defect detection are better than both TSR and PCT methods. The geometric characteristics of round and trapezoidal defects are clearly displayed in IV1. In IV2, the deepest diamond defects are revealed. Other IVs do not show useful information. In addition, the MLT method cannot display all the defects in one IV, which has caused inconvenience to the inspection process.

On a computer with 16 G of RAM, an Intel Core i7 CPU, and a Windows 7 operating system based on the TensorFlow framework, the GMLT method is calculated and implemented. The batch size is set to 5, the loss function is optimized using the Adam optimizer, and the learning rate is set at 0.002 during the SNGAN training. The training parameters of ISOMAP are: the number of neighbors is also 5 and the target dimension is 6.

It is inevitable to consider how much thermogram generation is appropriate in the GMLT method. Too few generations are not conducive to the learning of the model, whereas too many generations will not endlessly improve the model’s performance. In this work, considering that there were 90 raw thermal images, 80 simulated thermal images were generated. This is not guaranteed to be optimal, but after several tests, it was found that the performance of the GMLT method is relatively robust when varying around this quantity. Eventually, the whole dataset contains 170 thermal images. Figure 9a shows several generated thermal images. Comparing Figure 5 and Figure 9a, the original thermograms and the simulated thermograms are visually similar. Both of them contain severe inhomogeneous backgrounds from uneven heating and measurement noise. Defective information is masked so that detection becomes difficult.

As shown in Figure 9b, t-distributed stochastic neighbor embedding (t-SNE) [33] is used to visualize the distributions of the generated and original data. The marked numbers indicate the order of the original thermograms in the time sequence. It can be observed that the generated images are distributed around the original image in the middle of the sequence. A reasonable explanation is that the denoising and feature extraction ability of SNGAN make it more inclined to generate images similar to the information part of the training images [23]. In the original thermogram sequence, the first few frames contain a lot of noise and an uneven background, the last few frames contain less temperature contrast patterns, and the middle section contains the critical information that is the main object of learning for SNGAN. Therefore, the GMLT method incorporating the SNGAN data augmentation strategy has good defect detection performance.

The analysis results of the GMLT method are shown in Figure 10. It can be seen that a few IV thermograms can determine the geometric characteristics such as the shape and position of the defect. For details, IV1 extracts all the defect information, and their distribution locations are obvious. Additionally, IV2 clearly shows the boundary of each defect, which is more conducive to the identification of the defect shape. Compared with the first three methods, the GMLT method has better information separation and noise reduction, which promotes the reliability of IRT defect detection.

Table 1 compares the SNR values of several methods. Firstly, of the four methods, the proposed MLT and GMLT methods outperform in defect identification, because both have significantly larger SNR values than the other two methods. Secondly, it can be noted that the GMLT method has a significantly larger SNR value in all defects compared to the MLT method, which shows that the GMLT method better detects all defects. In addition, for individual defects, the GMLT method has a larger SNR value than the MLT method for diamond and circular defects, while the latter has slightly higher SNR values than the former for trapezoidal defects. In our opinion, such a slight difference in the SNR value indicates that the ratio of signal to noise in the visualized thermogram is similar for both methods.

The DBS values of several methods are listed in Table 2. It can be seen that GMLT has significantly the largest DBS values for both individual defects and all defects, indicating its best performance in separating defects and background information. In summary, the evaluation results of both metrics demonstrate that the GMLT method shows good IRT defect detection performance in composite materials.

### 4.3. Benefits Analysis of Thermogram Augmentation

The results in the previous section demonstrate that the GMLT method outperforms the MLT method in terms of defect identification, while the only difference between the two is the fused thermogram augmentation strategy or not. To explore the impact or benefit of data augmentation, the pixel distribution plots and probability density plots of MLT and GMLT in the directions of IVs are plotted, respectively.

Taking Figure 11a as an example, the top-right subplot shows the distribution of pixels in the IV1 and IV2 thermograms for the MLT method. The different colors and markers indicate the pixels in different regions of the thermal image. It shows that different classes of pixels are mixed together in a way that is difficult to distinguish. Therefore, the projections of the pixels in the IV1 and IV2 directions are plotted and used to show the separation of heterogeneous data. Although the projection map was intended to provide information on which IV direction the pixels are better separated in, it is still not obvious. In such a situation, the probability density plots are added to better show the separation of defect from the background (i.e., the pixels in the intact region are in one category and the pixels in the defective region are in the other category).

As shown in Figure 11a, the probability density plot of the data in IV1 shows that the data are better separated in that direction than in IV2. The DBS value in the lower left corner of the figure indicates that IV1 is a better projection direction than IV2, and the data can be better separated in this direction. As shown in Figure 7 for the MLT results, the defects of IV1 are indeed clearer than that of IV2. Compared with Figure 11a, the probability density plot of GMLT shown in Figure 11b on the same class of data on the IV1 projection is more distinguished, indicating that the GMLT method can maximize the separation of defect information from other information and achieve better defect detection results. Similarly, Figure 12 shows the advantages of GMLT and the importance of the projection direction with a probability density plot. Figure 12a shows that the MLT method has inferior data separation in both the IV3 and IV4 directions, which corresponds to the undetectable defect information in the IV3 and IV4 thermograms in Figure 8. In contrast, Figure 12b clearly shows that the data are almost inseparable in the IV3 direction of the GMLT, and the data are well separated in the IV4 direction, which is consistent with the defect information detected by IV3 and IV4 in the GMLT result shown in Figure 10.

In summary, the proposed GMLT method works well in the task of defect identification. It demonstrates that thermogram augmentation is an effective way to improve the performance of thermography analysis methods. A reasonable interpretation is that larger datasets allow the GMLT model to build a neighborhood graph that reflects the inherent structure of the data, and to obtain embedded components that retain its manifold after dimensionality reduction.

## 5. Conclusions

Many NDE applications of IRT require machine learning-based data analysis models to obtain faster and more accurate inspection results. The performance of existing thermographic data processing methods is often constrained by the limited thermogram dataset. In this work, an image augmentation strategy was considered to solve the problem of data shortage. A GMLT method was developed to promote the effectiveness of the IRT-based defect detection and assessment. This method employed a SNGAN-based augmentation strategy and PLSR to address the challenges of limited data and large display mapping solution errors in the manifold thermography method, respectively. The feasibility and superiority of the proposed method for detecting subsurface defects in CFRP specimens were experimentally illustrated. The t-SNE visualizations and the DBS index denote that the proposed method finds a better projection direction for compressing nonlinear thermographic data and obtains the intrinsic data structure. The limitation of this study is that the time series properties of the thermograms were not considered during image augmentation. Therefore, there is still room for improvement in thermogram augmentation for further enhancing detection accuracy. In the future, the problem of how to find generated images in a more efficient and effective manner, such as using 3D GAN, will be further investigated.

## Figures and Tables

**Figure 1 polymers-15-00173-f001:**
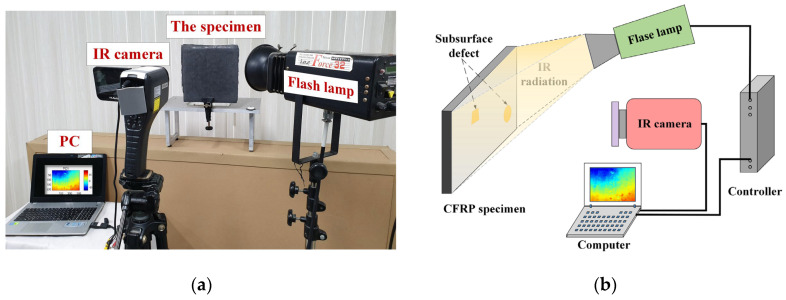
Flash thermographic NDT configurations: (**a**) experimental set-up; (**b**) schematic set-up.

**Figure 2 polymers-15-00173-f002:**
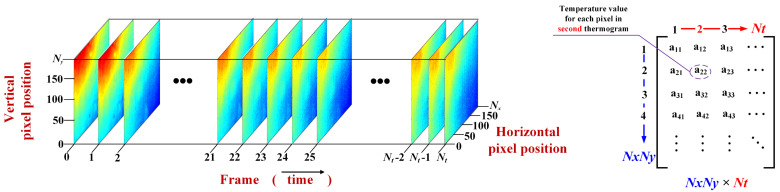
The 3D thermographic data sequence and 2D raster-like matrix.

**Figure 3 polymers-15-00173-f003:**
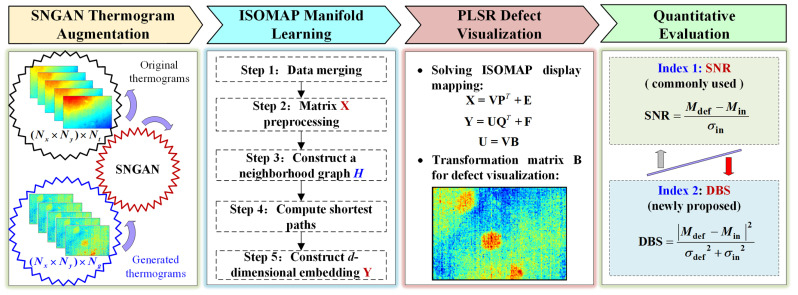
The GMLT framework and analysis procedure for thermographic data.

**Figure 4 polymers-15-00173-f004:**
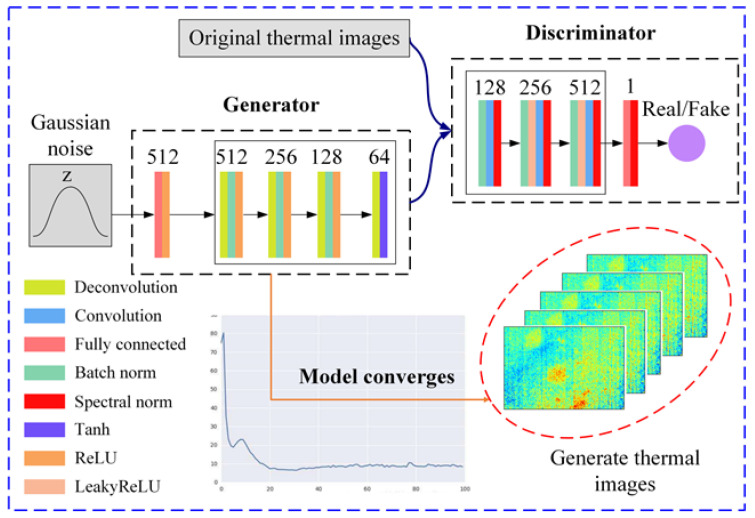
SNGAN data augmentation architecture for thermal image generation.

**Figure 5 polymers-15-00173-f005:**
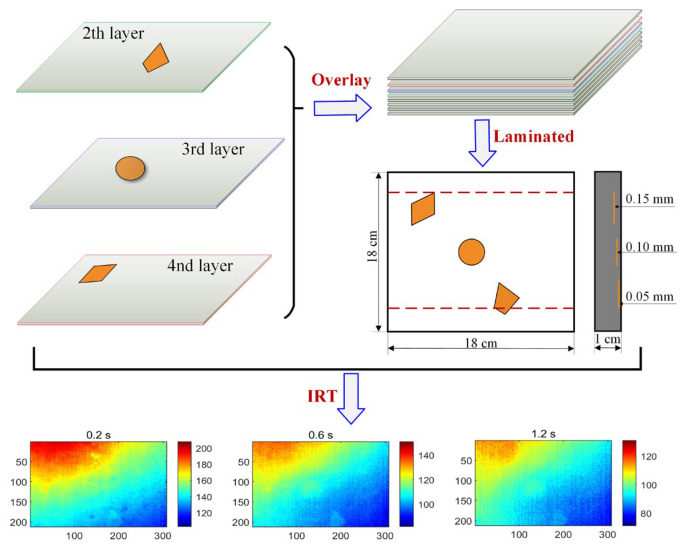
Fabrication of CFRP specimen with defects and several frames of raw thermal images.

**Figure 6 polymers-15-00173-f006:**
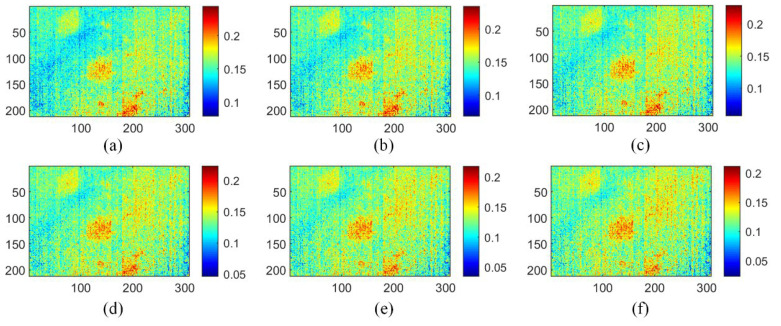
Reconstructed images of TSR method on three-defective CFRP sample: (**a**) first; (**b**) second; (**c**) fifteenth; (**d**) thirtieth; (**e**) sixty-second; (**f**) eightieth.

**Figure 7 polymers-15-00173-f007:**
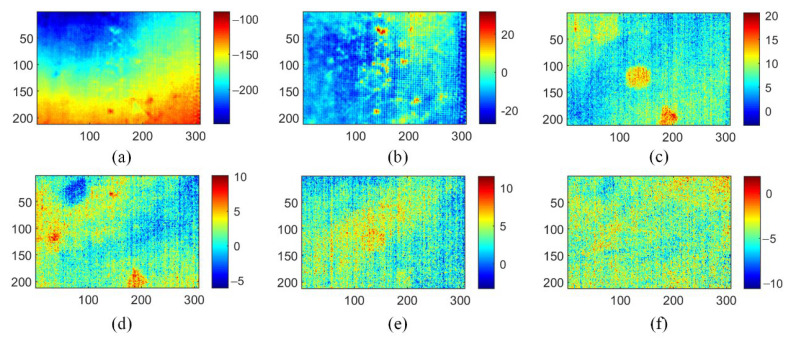
Analysis results of PCT method on three-defective CFRP sample: (**a**) PC1; (**b**) PC2; (**c**) PC3; (**d**) PC4; (**e**) PC5; (**f**) PC6.

**Figure 8 polymers-15-00173-f008:**
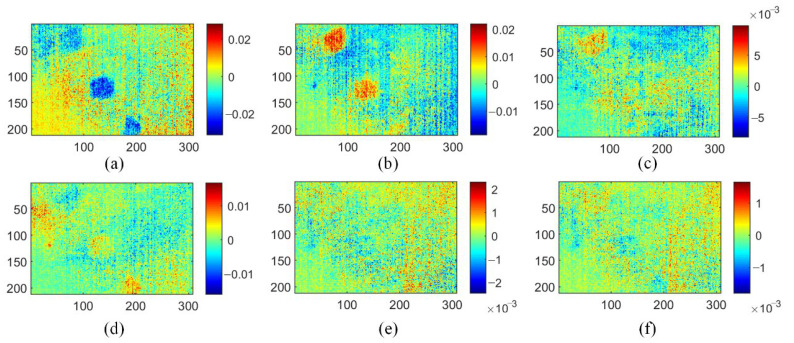
Analysis results of MLT method on three-defective CFRP sample: (**a**) IV1; (**b**) IV2; (**c**) IV3; (**d**) IV4; (**e**) IV5; (**f**) IV6.

**Figure 9 polymers-15-00173-f009:**
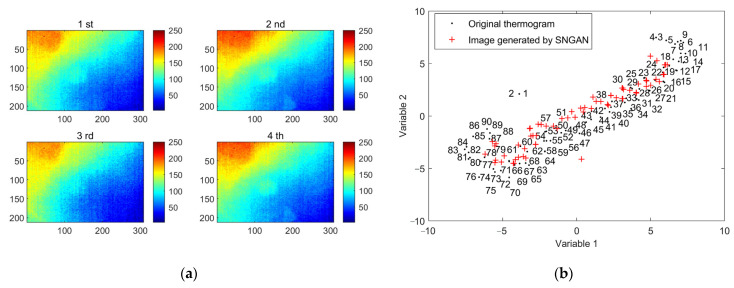
The SNGAN-generated thermal image data: (**a**) several simulated thermal images generated by SNGAN; (**b**) t-SNE visualization of raw and generated data.

**Figure 10 polymers-15-00173-f010:**
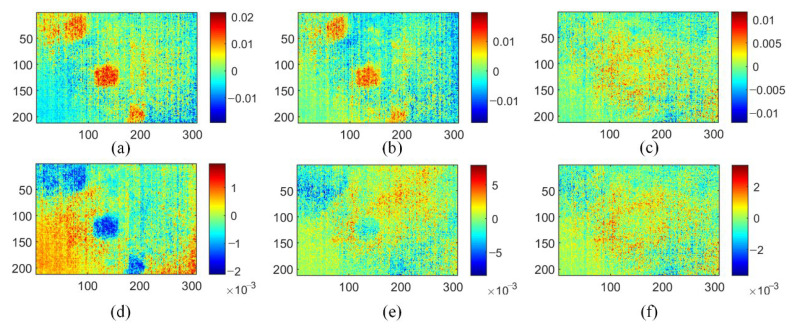
Analysis results of GMLT method on three-defective CFRP sample: (**a**) IV1; (**b**) IV2; (**c**) IV3; (**d**) IV4; (**e**) IV5; (**f**) IV6.

**Figure 11 polymers-15-00173-f011:**
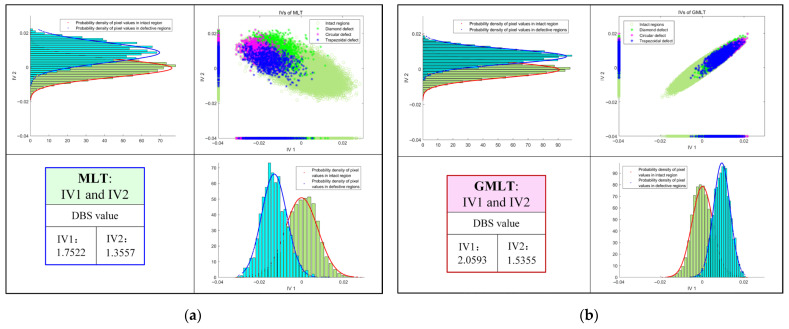
Pixel distribution and respective projections and their probability density plots in IV1 and IV2: (**a**) MLT; (**b**) GMLT.

**Figure 12 polymers-15-00173-f012:**
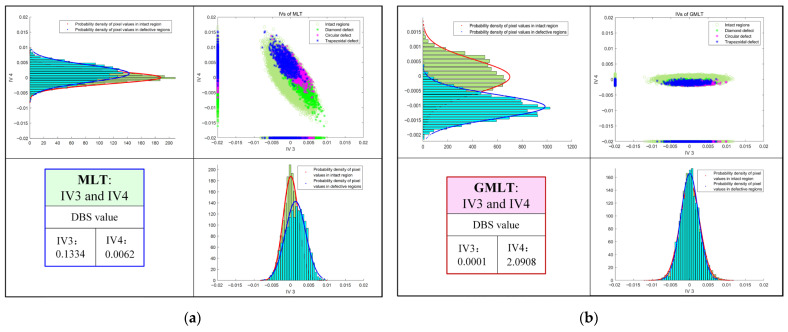
Pixel distribution and respective projections and their probability density plots in IV3 and IV4: (**a**) MLT; (**b**) GMLT.

**Table 1 polymers-15-00173-t001:** Comparison of SNR values of various methods.

	SNR			
	Diamond defect	Circular defect	Trapezoidal defect	All defects
TSR	0.64	1.61	1.39	1.32
PCT [11]	1.98	2.01	3.81	1.78
MLT [20]	2.33	2.50	4.21	2.35
GMLT	2.64	2.83	4.19	2.64

**Table 2 polymers-15-00173-t002:** Comparison of DBS values of various methods.

	DBS			
	Diamond defect	Circular defect	Trapezoidal defect	All defects
TSR	0.30	1.61	2.07	0.86
PCT [11]	1.22	2.29	2.91	1.01
MLT [20]	1.63	3.76	3.37	1.75
GMLT	2.18	3.92	3.78	2.09

## Data Availability

The data presented in this study are available on request from the corresponding author.

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
