# Peer review of "Data-Augmented Manifold Learning Thermography for Defect Detection and Evaluation of Polymer Composites"

_polymers, 2022, doi:10.3390/polym15010173_

Round 1
Reviewer 1 Report
Paper deals with important task. The authors proposed a novel generative manifold learning thermography approach for defect detection and evaluation of composites.
Paper has scientific novelty and great practical value.
Suggestions:
1. The introduction section should be extended using other data augmentation approaches. For example «input doubling method» and «additive input doubling method».
2. It would be good to add clear point-by-point the main contributions at the end of the Introduction section
3. The authors should provide a link to open access repository with the dataset used for modeling
4. It would be good to argue a choise of performance indicators used for evaluation. Why only two metrix?
5. The conclusion section should be extended using: 1) numerical results obtained in the paper; 2) limitations of the proposed approach; 3) prospects for future research.
Reviewer 2 Report
Dear authors,
After carefully reading your manuscript on a novel generative manifold learning thermography (GMLT) framework to enhance performance of IRT non-destructive defect evaluation for carbon fiber reinforced polymer (CFRP) composites, here are some comments and suggestions:
1. The Introduction section should include also some arguments for choosing CFRP.
2. Lines 72-76 could be removed.
3. Please explain how the sample is different from the one used in ref [20] and sample 1 in ref [22]. Ref [20] and [22] are not mentioned in the experimental part.
4. Please argue more on the choice of the defect shapes.
5. Figure 6- caption should include details on the a-f images.
6. Figure 7, 8, 10- captions should be expanded. Please include more details. Figures and their captions should stand alone.
7. Conclusions are too general.
